# Environmental Impact Assessment of Flexible Package Printing with the "LUNAJET®" Aqueous Inkjet Ink Using Nanodispersion Technology

**Katsuyuki Kozake [1,\*], Tsuyoshi Egawa [1], Satoshi Kunii [1], Hiroki Kawaguchi [1], Toru Okada [2], Yushi Sakata [2], Manabu Shibata [2] and Norihiro Itsubo [3]**

1   R&D—Performance Chemical Research, Kao Corporation, 1334 Minato, Wakayama 640-8580, Japan; egawa.tsuyoshi@kao.com (T.E.); kunii.satoshi@kao.com (S.K.); kawaguchi.hiroki@kao.com (H.K.)

2   ESG Action Management Group, Kao Corporation, 1-14-10, Nihonbashi Kayabacho, Chuo-ku, Tokyo 103-8210, Japan; okada.toru1@kao.com (T.O.); sakata.yushi@kao.com (Y.S.); shibata.manabu@kao.com (M.S.)

3   Graduate School of Environmental and Information Studies, Tokyo City University, Yokohama 224-8551, Japan; itsubo-n@tcu.ac.jp

\*   Correspondence: kozake.katsuyuki@kao.com; Tel.: +81-73-426-8563

**Abstract:** The share of digital printing on flexible plastic packaging has been increasing rapidly in response to the market demand for agility in Japan. To meet all these challenges, our response is the aqueous inkjet ink "LUNAJET®". By combining aqueous pigment nanodispersions with precise interfacial control technologies, "LUNAJET®" can contribute to the rapid digitization of flexible package printing while, at the same time, improving the environmental performance. Our analysis includes an evaluation of the environmental impact due to the conversion from gravure printing with an analog press to digital printing using an inkjet printer with water-based inks. In addition, inventory analyses and characterizations were carried out. It is shown that a 75% reduction in $CO_2$ emissions and 33% reduction in VOC (volatile organic compounds) emissions can be expected, particularly in small-lot printing, where digital printing is most desirable. An environmental impact assessment was conducted in Japan, based upon the LIME3 (life cycle impact assessment method based on endpoint modeling version 3) approach. It was found that the waste reduction rate for aqueous inkjet printing, compared to analog printing, was as high as 57% for small-lot production, assuming a large variety of products; surprisingly, the reduction rate remained at 15%, even for long production runs. As the market rushes to embrace digital printing technologies for packaging, these results indicate that implementing inkjet printing using aqueous ink for flexible plastic substrates can reduce waste and decrease the environmental burden, both for short as well as long printing runs.

**Keywords:** life cycle assessment (LCA); inkjet; flexible package; volatile organic content (VOC); water-based ink; digital printing

## 1. Introduction

### 1.1. New Water-Based Ink LUNAJET® and Digital Inkjet Printing

Flexible packages are used in many household consumables and provide invaluable contributions to the quality of our daily life by improving the protection, storage, appearance, and identification of the packaged contents. Nonetheless, as well as for the end consumer, the environmental burden created by flexible packaging—from its manufacture through to its use and disposal—has come under increased scrutiny, as has been noted in previous reports [1]. In particular, the space taken up by plastic waste in landfills and its impact on fisheries and marine ecosystems [2], as well as other environmental compartments, has been widely recognized as a problem, driving the need to develop technologies to recycle and recover waste flexible packaging.

This has led companies to invest more in the package design, as it is regarded as a marketing opportunity. The printing on its packaging differentiates a product from the competition and helps to shape consumer opinions regarding the value of the product [3]. This investment includes surveys and other efforts to understand what design elements appeal to consumers, how this appeal differs in different locations and for different cultures, and how it can change over time. This has led to the geographic customization of packaging, as well as the shortening of the product life cycle for particular package designs. The need to quickly customize packaging to different locations and changing consumer tastes over time favors the flexibility of digital over analog printing [4]. For these reasons, on-demand printing using inkjet or electrophotographic printing, which are compatible with short print runs and continuous variation in packaging design, has attracted interest and gained an increasing share of packaging printing in recent years [5]. Inkjet printing is a particularly promising printing method, as it has a printing speed comparable to conventional analog printing, as well as high color gamut, accessible through its unique image-forming process. The marketing and communication features of printing on flexible packaging can be improved by conversion to digital printing technologies.

Historically, inkjet printing has required solvent inks or UV curing inks to achieve acceptable performance for printing on non-porous hydrophobic surfaces, such as flexible plastic packaging. Compared to aqueous inks, solvent and UV inks pose greater safety (e.g., fire), health, and environmental risks, due to the VOC and UV curing agents used in their formulations. In contrast, a newly developed aqueous ink, "LUNAJET®" [6], is a safe and environmentally benign water-based inkjet ink, which enables high-quality printing on flexible plastic packaging.

Environmental impact assessments are used by local and national governments, as well as international and non-government organizations, to quantify the harm or benefit to the environment caused by human activities. The final disposition of materials was considered, from procurement to manufacturing of finished packaging materials, also taking into consideration disposal and recycling, together with relevant regulations, as an extended producer responsibility [7,8].

The environmental impact assessment conducted by Coca-Cola in 1969, at Mid-West Laboratories in the U.S., is reputedly the first example of such an analysis carried out for packaging [9]. It was based on the assessment of the environmental impact of different containers and the materials used. Since then, many similar examples have been reported as tools for various product and material selection processes [10–12].

Published examples of environmental impact evaluations have applied various approaches, such as LCA (life cycle assessment) beginning from the product design stage [13]; LCA evaluation of packaging materials, forms, and usage conditions of the products [14]; impact evaluation considering upstream and downstream sides of the supply chain as a manufacturing industry [15]; and evaluation of the impact on global warming based on the carbon footprint [16].

Most of the published LCA evaluations in the printing industry relate to home and office printers. In particular, for printers employing thermal fixing methods such as electrophotography, LCA evaluations have been conducted for various print volumes, where the evaluation methods include the impacts from the product design stage at each company to the end use by customers [17,18]. In contrast, for industrial package printing, different printing methods are used for different packaging materials and, typically, an optimum printing method and an optimum ink type dominate the printing market for a particular substrate and industry. However, databases on ink composition and printing modes are insufficient for conducting LCA evaluations of industrial printing processes. In particular, there is extremely limited information on aqueous inks and inkjet printing. In IDEAv2 (Inventory Database for Environmental Analysis Ver. 2) [19], the available data only considers oil-based inks, while the data indexed by printing method are available for rotogravure, offset, and relief printing but not for inkjet printing. In Ecoinvent [20], data are available only for offset and laser printing; there are no data for ink alone or for

inkjet printing. In this study, we consider it useful to calculate the environmental load for water-based ink and inkjet printing, as there only exist a few databases.

The environmental impacts of ink-derived raw materials, processing methods, and the responsibility of ink suppliers have been considered since the 1990s [21–23]. As an example, an LCA evaluation, together with an environmental performance improvement study, has been conducted, beginning from the raw materials for an ink derived from soybean oil, through to the ink manufacturing and printing processes [24]. The VOC emissions during printing with this soybean-derived ink are very low: only 0.11 kg per functional unit. However, the environmental burden of this ink is large, as this is only 0.47% of the total emissions (23.9 kg) produced during the cultivation of the raw material and manufacture of the ink. In addition to VOC release, the environmental performance was also low in terms of eutrophication potential, aquatic toxicity potential, and water use effect potential. The effect of reducing the environmental burden of rotogravure printing by replacing oil-based inks with water-based inks has been reported [25,26]. Fukumori et al. conducted LCA evaluations for three cases—oil-based roto-gravure printing, aqueous rotogravure printing, and oil-based gravure printing with solvent combustion treatment—and reported that water-based rotogravure printing had the best environmental performance. However, the water-based ink used in this study contained 25% alcohol and required more energy input for drying than the oil-based ink. For this reason, resource consumption was reduced by only 10% approximately and, in addition, a large amount of alcohol was released as a VOC into the atmosphere [26]. The printing industry awaits further improvements to reduce the environmental burden from ink manufacture and printing processes.

Many efforts have been made to reduce the environmental burden of printed substrates, and extensive research has also been directed toward the development of new packaging plastic film materials using biomass [27,28] or replacing a plastic substrate with paper having high biodegradability and low environmental burden at the time of manufacture [29,30]. However, the best currently available eco-conscious materials have diminished barrier, strength, and durability compared to conventional plastic film materials. If adopted, these packaging deficiencies could reverse the progress made towards reducing food and freshness loss due to the high functionality of flexible plastic packaging materials [31]. Such losses in the food industry impose their own environmental burden. LCA evaluations focusing on the food itself having a longer shelf life, rather than the packaging materials (e.g., biomass films), have also been carried out [32], and there are other questions related to the utility of biomass film materials and the functions required for food packaging [15,33]. Reducing the environmental burden of flexible plastic packaging requires consideration of the need to satisfy the requirements for packaging in each field of use. Companies developing new packaging material technologies must weigh their environmental impacts against their suitability for use.

Furthermore, environmental impact assessments of printing methods have been conducted in various fields, which have suggested that the $CO_2$ derived from raw materials used in the procurement of substrates accounts for over 60% of total emissions in the printing of papers and films [26]. The greatest improvement can be gained by eliminating the waste of ink and paper, film substrates, or proofing sheets. Digital printing can reduce the proofing sheets required for plate adjustment, ink color adjustment, and the operational losses that are associated with conventional analog printing. As a result, it is expected that one of the largest environmental burdens can be reduced by reducing the waste. Direct comparisons between different ink types have also been carried out using the same printing method [34]. Additionally, the environmental performance advantages that are unique to digital printing should be evaluated, such as short delivery time, printing of variable data, and reduction of loss, compared to the analog commercial printing [35]. However, an environmental impact assessment that has quantified and directly evaluated the differences between analog and digital printing methods, in addition to the environmental impact caused by materials (e.g., film and ink composition) related to direct printing, has not yet been reported. In this paper, we draw a direct comparison in the market in Japan between

the environmental performance of package printing using a rotogravure printing method with oil-based ink and package printing using an inkjet printing method with water-based ink. In performing the LCA, an inventory analysis and impact assessment were conducted for the procurement, production, and disposal under different printing length scenarios, including small lot size (which is more adaptable to customization) and longer printing runs in accordance with the international standard for ISO 14040 [36].

*1.2. Technology of the LUNAJET® Ink and the Inkjet Printer for Flexible Packages*

Although there exist various forms of flexible packaging, we describe a manufacturing process using a laminated packaging material that is often used for packaging food products as an example. The manufacturing process consists of the following three steps: (1) printing the design on one plastic film, (2) bonding this first plastic film to a second plastic film (with the printed surface between them) using a process called lamination, and (3) bagging by heat sealing. This chapter details Step (1)—the process of printing a design onto a film.

1.2.1. Challenges in the Printing Process

The first use of aqueous inkjets was for printing on porous (liquid-absorbing) substrates, such as paper, at home or in office sites. More recently, they have also become widely adopted in the commercial printing market for these types of substrates. In contrast, aqueous inkjet industrial printing has not yet significantly penetrated the market for printing on non-porous (non-liquid absorbing) substrates, such as flexible plastic packaging [37]. In general, aqueous inks do not penetrate these plastic film substrates. The drying of an aqueous ink printed on a porous substrate is assisted by a process called imbibition, wherein the ink fluid is wicked (drawn by capillary conduction) into the porous substrate, assisting in the drying process. When a conventional water-based ink designed for inkjet printing on paper stock is printed onto a plastic film substrate, the ink droplets spread and mix together on the surface of the film instead of penetrating, and the rate of drying is dramatically slowed down. This results in inter-color bleeding, wherein the ink droplets mix on the surface of the substrate, as well as causing the ink to remain wet and un-fixed on the substrate for a longer time (Figure 1).

**Figure 1.** Inter-color bleeding and approach to improvement of film printing challenges. [38].

Preventing such bleeding (color mixing) is the greatest challenge. If the pigment concentration is increased, in order to deliver a larger quantity of pigment in a smaller amount of liquid and enhance the drying property of the ink, the bleeding due to color mixing is improved (i.e., decreased). However, at this higher pigment concentration, the dot diameter of the ink on the film typically does not expand to an appropriate size, resulting in a deterioration in image quality, such as low density of an image. To solve this problem, it is

necessary to design a high-pigment concentration ink, which appropriately expands the ink dot diameter while also suppressing color mixing at the ink interface. Therefore, using our pigment nanodispersion [38] and precise interfacial control technologies [39] to optimize the ink properties, including the surface tension, high-quality flexible packaging printing using a pigment ink for an aqueous inkjet has been realized [6]. These two technologies are summarized below.

### 1.2.2. Pigment Nanodispersion Technology

Our pigment nanodispersion technology disperses and stabilizes the pigment at a nano size (about 100 nm) by uniformly coating the pigment surface with a unique functional polymer and dispersion process, designed by Kao (Figure 2). The pigment nanodispersion was prepared by a "high-pressure homogenizer". The pigment mixed with the polymer solution was dispersed under a pressure of 180 MPa using a Microfluidizer "High-Pressure Homogenizer" available from Microfluidics Corporation by passing the dispersion through the device 10 times or more to reduce the size of pigment powder to the nano size.

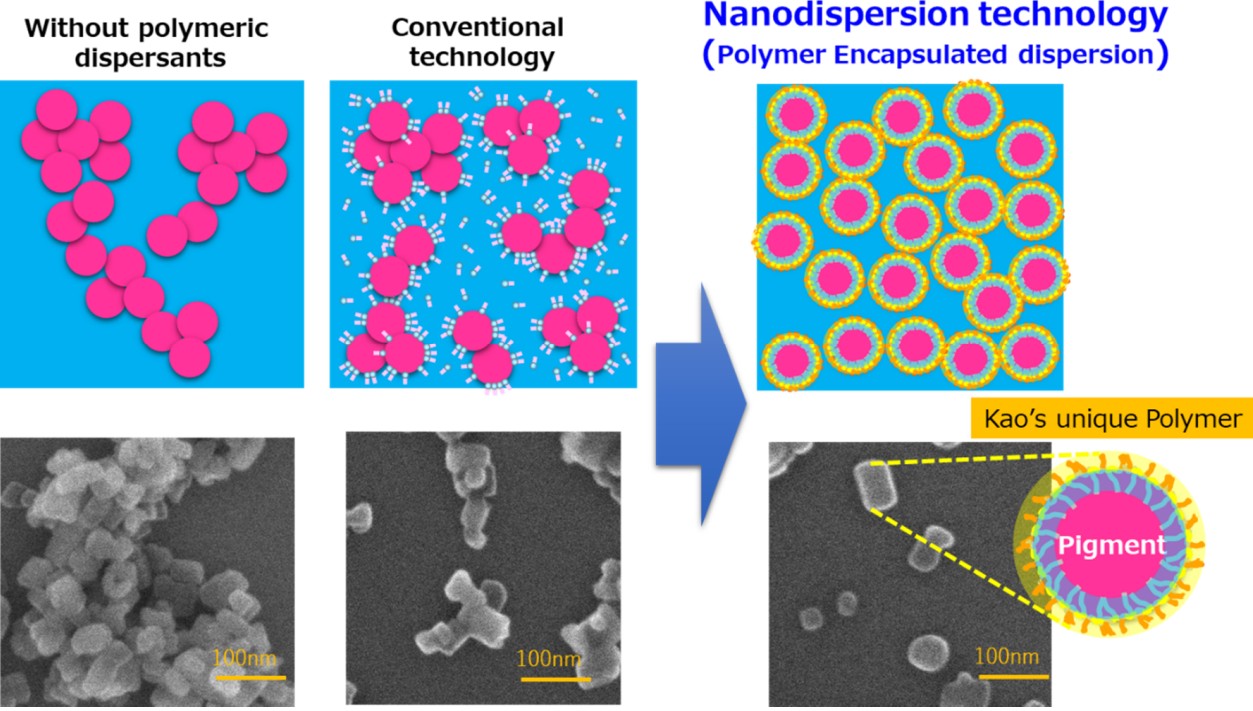

**Figure 2.** Concept of pigment nanodispersion technology.

When the particle size of a pigment powder (aggregate of primary particles) is reduced by milling into (nano-size) primary particles in water, the particle size reduction is accompanied by an increase in the hydrophobic pigment surface area exposed to water. The unfavorable interaction between the increased hydrophobic pigment surface area and water molecules results in increased free energy ($\Delta G$). In the absence of additional steps taken to thermodynamically stabilize this increased surface area, the pigment particles will re-agglomerate over time, once again decreasing the surface area exposed to water and lowering the free energy $\Delta G$ [40].

Functional polymeric dispersants can stabilize the interface between pigments and water. They incorporate hydrophobic domains that are able to drive a favorable association between the dispersant and the hydrophobic surface of the pigment, along with hydrophilic domains that include electrostatic repelling groups (e.g., -COO·X+; X = Alkali metal ions) and steric repelling groups (e.g., ether chains), which are able to favorably associate with water. As any two dispersed particles approach each other, the concentration of ions and polyether chains from the dispersant increases locally between the particles. This

results in a countervailing force (i.e., osmotic effect [41]), which tends to drive the particles apart, diluting this concentration and preventing the aggregation of the two particles, thus allowing them to maintain a dispersed state. Using this technique, premature aggregation is suppressed in the drying step of the ink on a film, and the ink dots are able to spread to an appropriate size. Therefore, compared to an ink prepared without this pigment nanodispersion technique, the novel ink contributes to a reduction in the amount of ink required for printing (which was 12% in an example). In the presented LCA evaluation, the ink amount after reduction is used.

### 1.2.3. Precise Interfacial Control Technology

In the actual printing process, as printing is performed for each color, there is a delay between the times when ink droplets of different colors reach the substrate. Evaporation during this time difference creates a difference in surface tension between the droplets. When ink droplets with different surface tensions are adjacent to each other, the ink droplets with lower surface tension tend to wet the ink droplets with higher surface tension. This phenomenon causes the inks to spontaneously mix. Therefore, in order to suppress color mixing, it is necessary to precisely control the physical properties of the ink. Inks incorporating our technologies to improve drying and control the physical properties lead to improved print quality by suppressing blurring due to inter-color bleeding compared with competitive ink designs.

### 1.2.4. Summary of Ink Design Technologies for the Printing Process

By developing improved technologies and designing pigment inks for an aqueous inkjet—that is, using our novel pigment nanodispersion and ink physical property control technologies—it was possible to realize high image quality using aqueous pigment inks on non-porous substrates. In March 2016, the company made a press release, with the title "Realize film printing for flexible packaging using the world's first ink technology".

With the technological advancement of printing machines and inks, digital printing and aqueous inkjets are progressing in the industrial printing market for non-porous substrates, such as flexible plastic package printing. We are convinced that a new era of industrial printing has begun.

## 2. Materials and Methods

### 2.1. Functional Units and System Boundaries

#### 2.1.1. Targeted Products

For this study, we compiled and evaluated data for two different print run lengths (21,000 m$^2$ and 2100 m$^2$) on flexible packaging, for each of two different printing technologies (oil-based ink gravure printing versus water-based inkjet printing), in order to compare the estimated environmental burden of each printing method for both short and long print runs (cumulatively, four different printing scenarios). In all four scenarios, the multi-layer flexible package made of polypropylene with a capacity of 340 m, sold by the Kao Corporation, Japan, was used as the model substrate.

#### 2.1.2. Evaluation Scenario

The four scenarios below were evaluated and compared. (Table 1).
Scenario 1: Oil-based ink/gravure printing method/21,000 m$^2$ printing.
Scenario 2: Oil-based ink/gravure printing method/2100 m$^2$ printing.
Scenario 3: Water-based ink/inkjet printing method/21,000 m$^2$ printing.
Scenario 4: Water-based ink/inkjet printing method/2100 m$^2$ printing.

**Table 1.** Scenario comparison.

| No. | Name | Contents |
|---|---|---|
| Scenario 1 | Analog printing<br>Oil-based ink/gravure printing system<br>21,000 m$^2$ Print<br>(Oil GR-L) | Inks: 6 colors (oil-based), printing plates: 6<br>Ink application: 5.4 g/m$^2$<br>Film length: 20,000 m, film width: 1.05 m<br>Power for printing: 270 kWh<br>Film loss at startup: 210 m$^2$ (200 m)<br>Ink loss: 6 kg/color (CMYK), 10 kg/color (special color)<br>Ink volatile component: atmospheric emission<br>Film loss and ink loss: incineration (including heat recovery) |
| Scenario 2 | Analog printing<br>Oil-based ink/gravure printing system<br>2100 m$^2$ Print<br>(Oil GR-S) | Inks: 6 colors (oil-based), printing plates: 6<br>Ink application: 5.4 g/m$^2$<br>Film length: 2000 m, film width: 1.05 m<br>Power for printing: 29 kWh<br>Film loss at startup: 210 m$^2$ (200 m)<br>Ink loss: 6 kg/color (CMYK), 10 kg/color (special color)<br>Ink volatile component: atmospheric emission<br>Film loss and ink loss: incineration (including heat recovery) |
| Scenario 3 | Digital printing<br>Water-based ink/inkjet printing system<br>21,000 m$^2$ Print<br>(Water IJ-L) | Inks: 6 colors (Water-based), no printing plate<br>Ink application: 4.8 g/m$^2$<br>Film length: 20,000 m, film width: 1.05 m<br>Power for printing: 565 kWh<br>Film loss at startup: 21 m$^2$ (20 m)<br>Ink loss: 1 kg/color (CMYK, special color)<br>Ink volatile component: atmospheric emission<br>Film loss and ink loss: incineration (including heat recovery) |
| Scenario 4 | Digital printing<br>Water-based ink/inkjet printing system<br>2100 m$^2$ Print<br>(Water IJ-S) | Inks: 6 colors (Water-based), no printing plate<br>Ink application: 4.8 g/m$^2$<br>Film length: 2000 m, film width: 1.05 m<br>Power for printing: 57 kWh<br>Film loss at startup: 21 m$^2$ (20 m)<br>Ink loss: 1 kg/color (CMYK, special color)<br>Ink volatile component: atmospheric emission<br>Film loss and ink loss: incineration (including heat recovery) |

In "Oil-based Gravure 21,000 m$^2$ Printing" (hereafter, "Oil GR-L") in Scenario 1, six colors of oil-based inks and six printing plates were used, and 21,000 m$^2$ (20,000 m $\times$ 1.05 m) were printed, using the gravure method, onto plastic film substrates made of polypropylene. There are various plastic film substrates used in Japan such as polyethylene, polyethylene terephthalate, nylon as polyamide, and polypropylene. However, polypropylene was used in this study as a representative material since polypropylene is one of the most popular plastic film substrates among them [42,43]. In this case, the amount of ink applied was set to be 5.4 g/m$^2$ from the actual value. The film loss at the start-up of the print was 210 m$^2$ (200 $\times$ 1.05 m); the ink loss was 6 kg for the four colors of CMYK (the reference colors) and 10 kg for the other special colors; and these losses were incinerated (including thermal recovery). Electric power was used for both the printer and the ink dryer, and the volatile ink components were discharged into the atmosphere during drying.

Scenario 2, "Oil-based Gravure 2100 m$^2$ Printing" (hereinafter referred to as "Oil GR-S"), was a scenario in which the print area of Scenario 1 was changed to 2100 m$^2$ (2000 $\times$ 1.05 m), while the other features were the same as in Scenario 1.

In "Water-based Inkjet 21,000 m$^2$ Printing" (hereafter, "Water IJ-L") in Scenario 3, six colors of water-based inks were used, and 21,000 m$^2$ (20,000 $\times$ 1.05 m) were printed, using an aqueous inkjet method, onto plastic film substrates made of polypropylene. In this case, the ink application rate was 4.8 g/m$^2$, from the actual values, and the ink could be printed more efficiently than the oil-based ink, through the use of the "LUNAJET®"

aqueous pigment ink technology. The film loss at the time of printing startup was 21 m$^2$, the ink loss was 1 kg per color, and the other features were the same as in Scenario 1.

In "Water-based Inkjet 2100 m$^2$ Printing" (hereinafter "Water IJ-S") in Scenario 4, the print area of Scenario 3 was changed to 2100 m$^2$ (2000 × 1.05 m). The other features were the same as in Scenario 3.

### 2.1.3. Functional Unit

Two scenarios—21,000 m$^2$ (equivalent to 200,000 flexible packages) and 2100 m$^2$ (equivalent to 20,000 flexible packages, as the minimum lot area of general gravure printing)—were evaluated, focusing on the printing processes of Oil GR and Water IJ. The functional unit was one flexible package (0.105 m$^2$).

### 2.1.4. System Boundaries

The life-cycle flow and system boundaries for each scenario are shown in Figure 3. The system boundary was the printing process of the flexible package, and the calculation was carried out for the raw material procurement (ink procurement, ink transport, material procurement), printing, and disposal stages; the subsequent stages were excluded.

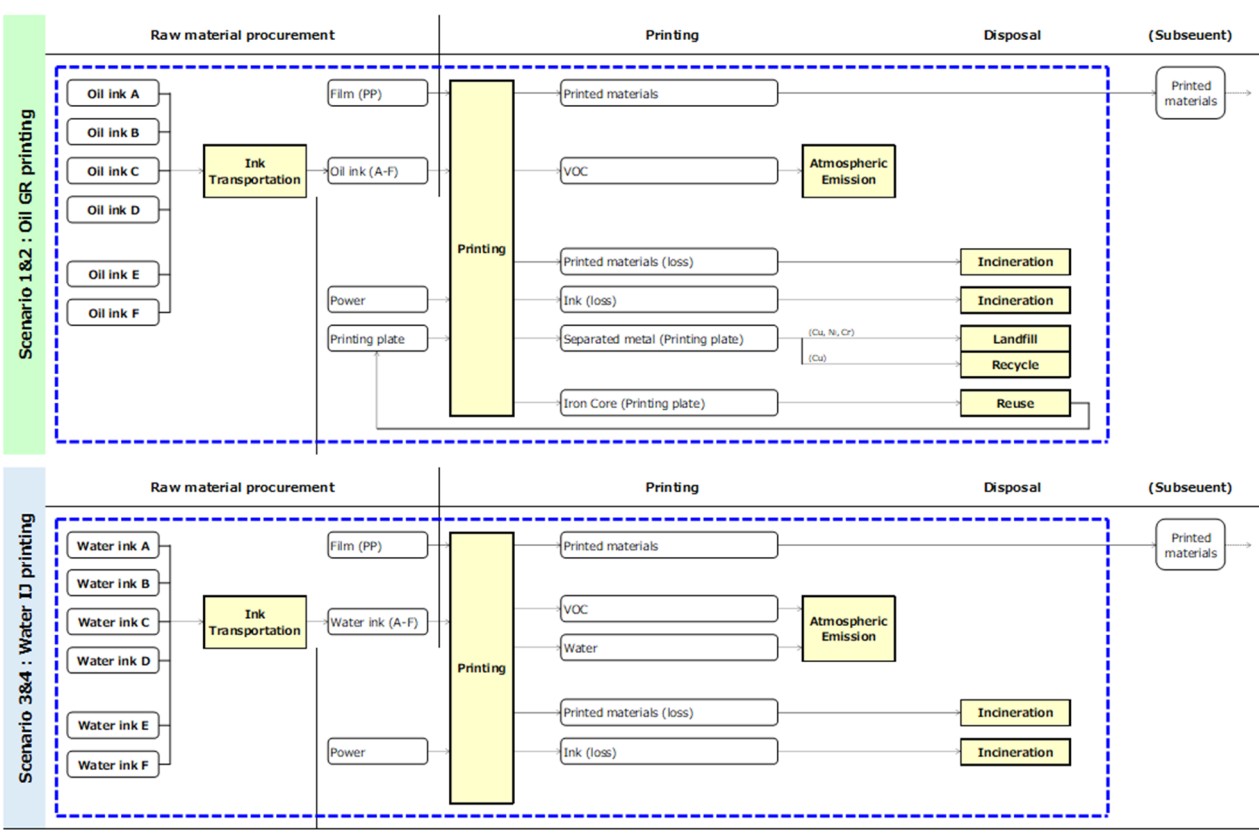

**Figure 3.** Life-cycle flow and system boundary of each evaluation system. (The range surrounded by the blue line is defined as the system boundary. The environmental loads of the printer device production were not included in the calculation boundary in this study).

### 2.2. Data Collection

The database in Japan was used in this study because it assumed a scenario in Japan. The database list used in this study was shown in attached Table 2.

**Table 2.** Representative databases used in this study.

| Stage | Process/Flows | | Database | | Remarks |
|---|---|---|---|---|---|
| Raw material procurement | Ink | Pigments | IDEAv2 | Ring intermediates, synthetic dyes, organic pigments | |
| | | VOC | IDEAv2 | Ethyl acetate | representative |
| | | Water | IDEAv2 | Pure water, ion exchange membrane method | |
| | Ink transportation | | IDEAv2 | Trucking service, 10-ton vehicle, 75% load rate | 485 km |
| | Film | | IDEAv2 | Plastic film production | |
| Printing | Power | | IDEAv2 | Tokyo Electric Power, 2015 | |
| | Printing plate | Cu | IDEAv2 | Crude copper production | |
| | | Ni | IDEAv2 | Metal nickel production | |
| | | Cr | IDEAv2 | Chromic acid production | |
| Disposal | Incineration | Ink | IDEAv2 | Incineration services, industrial waste, waste oil | |
| | | Printed materials | IDEAv2 | Incineration services, industrial waste, waste plastics | |
| | Landfill | Cu, Ni, Cr | IDEAv2 | Industrial waste treatment, scrap metal | |
| | Recovered energy | Heavy oil | IDEAv2 | Production of heavy oil A | |

2.2.1. Raw Material Procurement Stage

For the ink materials, we used the composition of LUNAJET® sold by Kao Corporation to estimate the environmental burden for water-based ink and used the analysis results for oil-based gravure ink on the market. In the ink transport, a transport condition of 10-ton trucks and a loading ratio of 75% was assumed. The transport distance was set as 485 km, which is the distance between the ink production factory and the location of the printing supplier. The film material was evaluated assuming a plastic film (polypropylene; width, 1050 mm; thickness, 20 μm), which is generally used for printing. Inventory data for ink materials, ink transportation, and film materials were obtained from IDEAv2.

2.2.2. Printing Stage

In the printing stage, power is used to run the printing press and to dry the ink. In terms of power quantities, that for water-based inkjet printing used data from similar production lines, while that for oil-based gravure printing used measured data from in-house gravure printers. In addition, VOCs discharged by ink drying at the time of printing were assumed as atmospheric releases. For the printing plates used for gravure printing, the process of applying metal plating to the iron core was modeled through interviews with suppliers. IDEAv2 values were also used for power and inventory data of the metals used in the plate.

2.2.3. Disposal Stage

Ink and printed material losses were assumed to be disposed of by incineration, thermal energy at the time of incineration was also assumed to be 100% recovered, and recovered energy value was calculated using heavy oil inventories, which are obtained from IDEAv2. The printing plate was considered as a scenario in which the plated metal was separated from the iron core, the iron core was reused, and the separated metal was further fractionated using an eluent and then landfilled (Cu, Ni, and Cr) and recycled (Cu). The ratios of landfill and recycling were 64% and 36%, respectively, depending on the primary data obtained from interviews results with suppliers. There are various proposals

reported for collection and recycling [44,45]. In this study, the calculation was based on subtracting the input amount of primary data instead of the allocation fractionated and recycled ratio. IDEAv2 was also used for inventory data on the incineration of wastes, landfill of metals, energy during incineration, and procurement of heavy oil, which were used for calculations regarding the disposal stage.

### 2.3. Environmental Impact Assessment (Characterization) Method

LIME2 (life cycle impact assessment method based on endpoint modeling version 2) [46] was applied for characterization of environmental impact assessment.

### 2.3.1. Global Warming

The environmental impact assessment of global warming was carried out using inventory analyses of LC-$CO_2$, LC-$CH_4$, and LC-$N_2O$. The characterization was evaluated by $CO_2$ conversion (kg-$CO_2$ eq.) using the AR5 coefficients of IPCC ($CO_2$ = 1, $CH_4$ = 28, and $N_2O$ = 265).

### 2.3.2. Photochemical Oxidants

The environmental impact assessment of photochemical oxidants was carried out using the inventory analyses of LC-$NO_X$ and LC-NMVOC. The characterization was carried out in ethylene equivalent (kg-ethylene eq.), using the coefficients of LIME2.

### 2.3.3. Fossil Fuel Consumption

The results of inventory analyses of LC-Oil, LC-Coal, and LC-NG were used to assess the environmental impact of fossil fuel use. The characterization was evaluated by energy conversion (MJ), using the coefficients of LIME2.

### 3. Results

### 3.1. Results of Inventory Analysis

We show representative results for the inventory analysis of each scenario such as $CO_2$ emissions, NMVOC emissions, crude oil consumption, and natural gas consumption in Figures 4 and 5. Other inventory analysis results used for this study are shown in the attached.

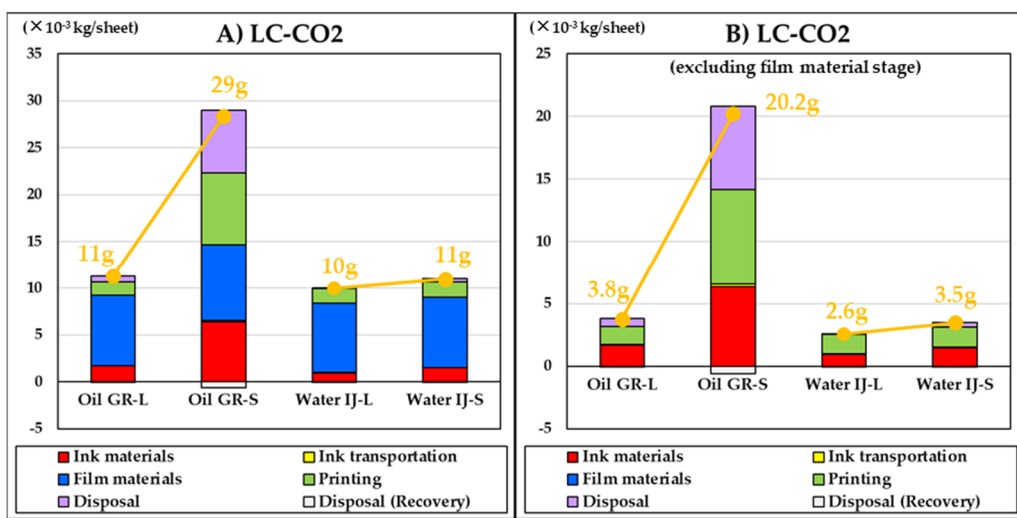

**Figure 4.** Inventory of $CO_2$ emissions: (**A**) entire printing process; (**B**) excludes the film material stage. Line graph is the net inventory plus heat recovery at disposal.

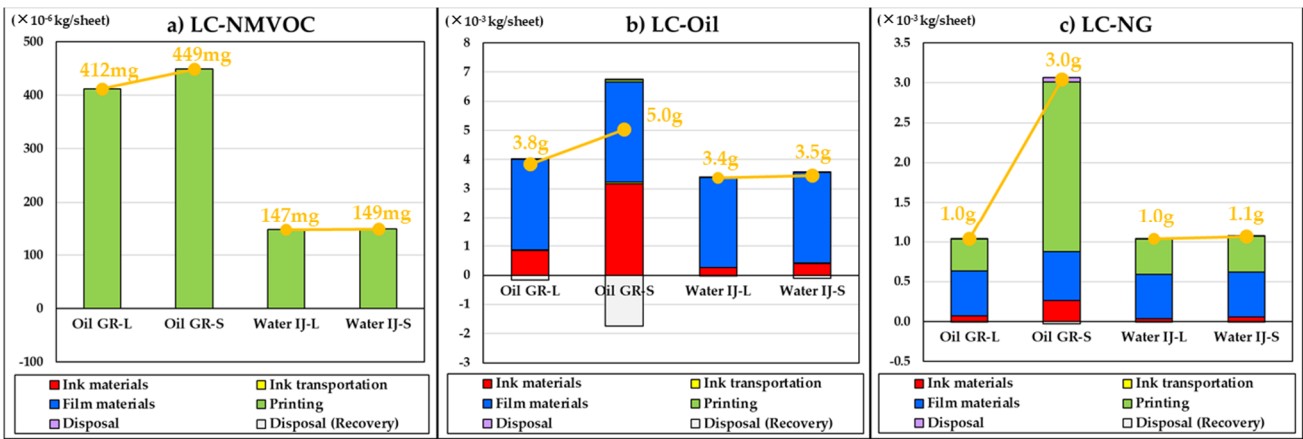

**Figure 5.** Inventory results: (**a**) NMVOC emissions; (**b**) crude oil consumption; and (**c**) natural gas consumption. (Line graph is the net inventory plus heat recovery at disposal).

### 3.1.1. Life Cycle $CO_2$ Emissions (LC-$CO_2$)

Figure 4 shows the analyses of life cycle $CO_2$ emissions. The net life cycle $CO_2$ emissions, minus the effects of heat recovery at disposal, were 11, 29, 10, and 11 g/sheet for the Oil GR-L, Oil GR-S, Water IJ-L, and Water IJ-S scenarios, respectively. In 21,000 m$^2$ prints of large lots, there was no major difference between the Oil GR and Water IJ printing; meanwhile, in the 2100 m$^2$ small lot prints, the $CO_2$ emission amount increased drastically for the Oil GR print, whereas, for the Water IJ print, there was only a small change. This was caused by the difference in the amount of ink loss, due to the difference in the printing method. The ink loss for each print run was set to 6 kg in total for Water IJ printing and 44 kg in total for Oil GR printing, regardless of the printing area, with the large loss for gravure printing attributed to the use of an ink pan. Therefore, we estimated that the $CO_2$ emissions due to the ink raw material per sheet were increased for small-lot printing for the Oil GR, due to a large ink loss. Furthermore, regarding the ink composition, as a large amount of water was blended in the water-based ink, the $CO_2$ emissions due to the raw ink material were reduced. In addition, we considered that the application amount per unit area of the film was about 10% smaller (at 4.8 g/m$^2$) for the water-based inkjet ink compared to the Oil GR (estimated 5.4 g/m$^2$) ink use, which was accomplished through the improvement of the pigment dispersing technology.

The $CO_2$ emissions at the print stage for the Oil GR-S were about five times greater than in the other scenarios. This was attributable to the power consumed to manufacture the print plates, contributing a large load per sheet for small printing lots. Comparing the production stages of Oil GR-L to Water IJ-L, Oil GR-L resulted in smaller values, despite the loading of the printing plate. This was because the load of the printing plate per sheet decreases as the lot size increases, and the power used in the drying process at the time of printing in the Water IJ is larger than that in the Oil GR, as oil-based inks are composed largely of VOC, which are more easily volatilized, whereas water-based inks contain a large amount of water, which requires more power for drying. In LUNAJET$^®$, the droplet control forms droplets smaller than ordinary water-based inks, and the power required for drying is reduced, compared to competitive water-based inks, but more power is used than for oil-based inks. Due to the large ink loss, $CO_2$ emissions were drastically increased in the Oil GR-S, as well as in the ink procurement stage. In Oil GR-L, the ink loss per sheet became relatively small; therefore, the $CO_2$ emissions were comparable to those of Water IJ-S and the Water IJ-L.

Focusing on the ink material and the printing method only (which are the focus of this study), the results excluding the film material stage—which is a common base material and has a large load—are shown in Figure 4b. The net life cycle $CO_2$ emissions, minus the effects of heat recovery at disposal, were 3.8, 20.2, 2.6, and 3.5 g/sheet, respectively, for Oil GR-L, Oil GR-S, Water IJ-L, and Water IJ-S. Comparing the effects of different printing

methods in large-lot printing areas, Water IJ-L printing reduced the $CO_2$ emissions by about 32% compared to the Oil GR-L. Furthermore, even for Water IJ-S, where the printing efficiency is reduced by the smaller printing area, the $CO_2$ emissions were reduced by about 8% compared to oil GR-L.

### 3.1.2. Life cycle NMVOC Emissions (LC-NMVOC)

The analyses of life cycle NMVOC emissions (LC-NMVOC) are shown in Figure 5a. The net life cycle $CO_2$ emissions, minus the effects of thermal recovery at disposal, were 412, 449, 147, and 149 mg/sheet for the Oil GR-L, Oil GR-S, Water IJ-L, and Water IJ-S scenarios, respectively, resulting in a 60–70% reduction for aqueous inkjet printing, regardless of printing lot size. The loading of LC-NMVOC is mainly due to the volume of VOC generated during the drying process in the printing stage. Therefore, as the VOC content is low in the aqueous ink composition and the amount of ink applied per unit area is smaller in Water IJ printing, the result was a significantly smaller value for VOC emissions.

### 3.1.3. Life Cycle Crude Oil Consumption (LC-Oil)

The analytical results of the life cycle crude oil consumption (LC-Oil) are shown in Figure 5b. The net life cycle crude oil consumption, after deducting heat-recovery effects at disposal, was 3.8, 5.0, 3.4, and 3.5 g/sheet, respectively, for the Oil GR-L, Oil GR-S, Water IJ-L, and Water IJ-S scenarios, wherein the large value in the Oil GR-S scenario was attributed to large ink and print loss per sheet. Offsetting this was the fact that, in the Oil GR-S scenario, where ink and paper losses were large, more thermal energy was recovered when incinerated by fossil fuel A (as high as 1.7 g/sheet).

Comparing the large-lot print scenarios, the LC-Oil loading for Water IJ-L at the ink material stage was significantly lower—about 30%, compared to Oil GR-L. This was not surprising, as the amount of crude oil used to manufacture water-based inks is decreased significantly, due to the proportion of oil-derived components in the ink composition being lower and the ink quantity applied per unit during printing being smaller than that for oil-based inks.

### 3.1.4. Life Cycle Natural Gas Consumption (LC-NG)

The results of analyzing the life cycle natural gas consumption (LC-NG) are shown in Figure 5c. The net life cycle natural gas consumption, after heat recovery at disposal, was 1.0, 3.0, 1.0, and 1.1 g/sheet for the Oil GR-L, Oil GR-S, Water IJ-L, and Water IJ-S scenarios, respectively. Thus, Oil GR-S was about three times higher than other scenarios. Of this, approximately 70% was attributed to the printing stage, due to the electric power used in the metal plating process (i.e., the printing plate manufacturing process). On the other hand, in the case of Oil GR-L, the loading of printing plates per sheet was reduced, due to the increase in the number of sheets produced.

### 3.2. Characterization Results

The characterization results of global warming, photochemical oxidants, and fossil fuel consumption are shown in Figure 6.

### 3.2.1. Global Warming

The characteristics of global warming were 13, 39, 10, and 11 g-$CO_2$ eq./sheet for the Oil GR-L, Oil GR-S, Water IJ-L, and Water IJ-S scenarios, respectively, where $CO_2$ emissions dominated all the scenarios. In the oil GR scenarios, $CH_4$ emissions contributed 10–25%. This is thought to have resulted from the power used in the printing plate manufacturing. In all scenarios, the effect of $N_2O$ was small.

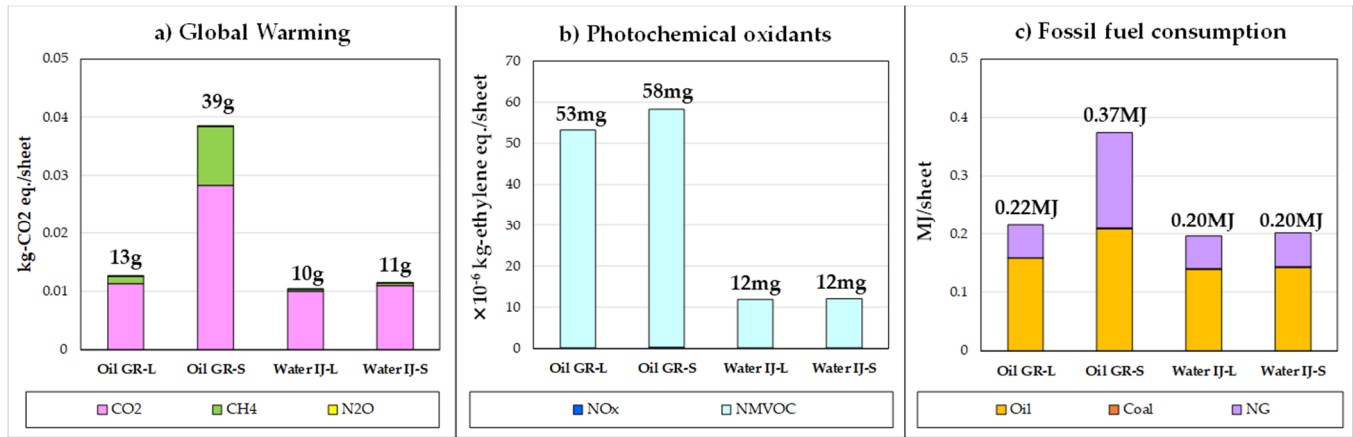

**Figure 6.** Characterization results: (**a**) global warming; (**b**) photochemical oxidants; and (**c**) fossil fuel consumption.

3.2.2. Photochemical Oxidants

The characterization results of photochemical oxidants were 53, 58, 12, and 12 mg-ethylene eq./sheet for the Oil GR-L, Oil GR-S, Water IJ-L, and Water IJ-S scenarios, respectively. The effect of $NO_x$ on photochemical oxidants was small, and most of it was the effect of NMVOC, because all volatile components of the ink are released into the atmosphere in this study. Inventory analyses showed that, for Water IJ printing, LC-NMVOC was reduced by 60–70%, compared to Oil GR printing, regardless of the printed area. However, taking into account the characterization of photochemical oxidants, these differences increased to around 80%. This was because many of the VOC in water-based inks are glycols (0.06 kg-ethylene eq./kg), the photochemical oxidant coefficient of which is smaller than those of esters (0.11–0.13 kg-ethylene eq./kg) and hydrocarbons (0.21 kg-ethylene eq./kg), which are the dominant VOC in oil-based inks.

3.2.3. Fossil Fuel Consumption

The characterization results for fossil-fuel consumption were 0.22, 0.37, 0.20, and 0.20 MJ for the Oil GR-L, Oil GR-S, Water IJ-L, and Water IJ-S scenarios, respectively. In all scenarios, the impact of crude oil was the largest, followed by a large share of natural gas, while the impact of coal was nearly absent. Oil GR-S was about 1.8 times higher than all other scenarios, as Oil GR-S consumed crude oil to make the ink and used electric power (derived from natural gas) to make the printing plates.

In addition, comparing Oil GR-L and Water IJ-L, the proportion derived from crude oil was smaller for Water IJ-L. This is because preparing water-based ink requires less energy derived from crude oil than oil-based ink; furthermore, the amount of ink per unit printed area is smaller. Although the loads derived from natural gas were equivalent, it was presumed that the electric power used in the drying process for the printing stage accounted for the natural gas consumption in Water IJ-L, whereas the electric power used in the printing plate manufacturing accounted for natural gas consumption in Oil GR-L.

## 4. Discussion

### 4.1. Environmental Impact of Ink Composition and Printing Method

We have reported a comparison of the environmental burden in the printing process per unit (refillable flexible plastic film package) printed, which was carried out for large-lot and small-lot printing scenarios using Water IJ and Oil GR printing systems. However, as both the ink composition and printing method are different for Water IJ and Oil GR scenarios, the cause of the difference in environmental burden was ambiguous. In this section, our purpose is to separate the contributions to the differences made by the ink composition from those made by the differences in the printing technology, by comparing Water IJ and Oil GR scenarios to an oil-based inkjet printing method (Oil IJ). The Oil IJ printing scenario is detailed in Table 3. In Figure 7, the outcomes of LC-$CO_2$ and LC-

NMVOC per flexible package for printed lots of Water IJs, Oil IJs, and Oil GR prints are also shown.

**Table 3.** Oil-based inkjet printing scenario.

| No. | Name | Contents |
|---|---|---|
| Scenario 5 | Digital printing Oil-based ink/inkjet printing system 1100–55,000 m$^2$ (Oil IJ) | Inks: 6 colors (oil-based), no printing plates Ink application: 5.4 g/m$^2$ Film length: 1000–50,000 m, film width: 1.05 m Film loss at startup: 21 m$^2$ (20 m) Ink loss: 1 kg/color (CMYK, special color) Ink volatile component: atmospheric emission Film loss Ink loss: incineration (including heat recovery) |

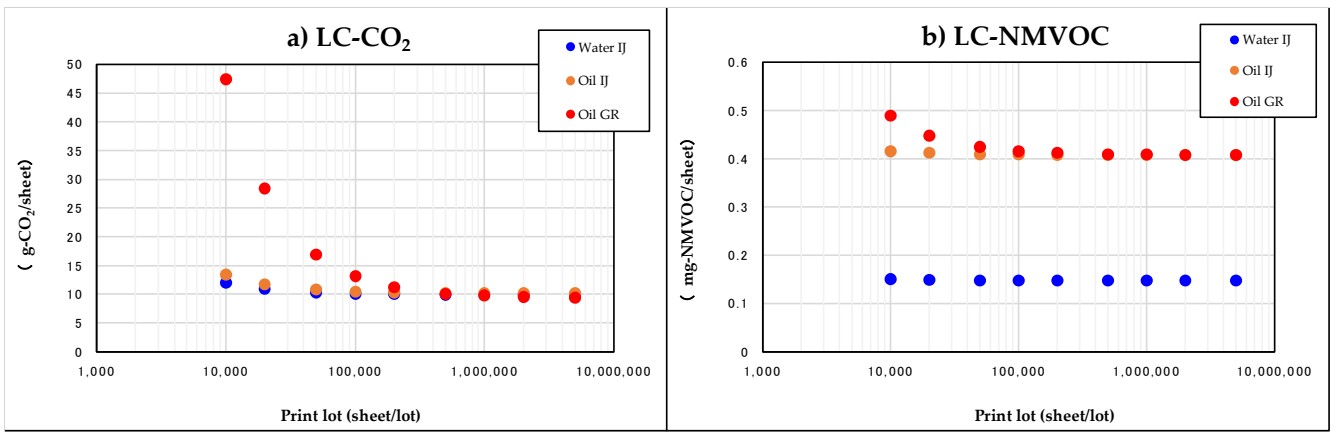

**Figure 7.** Sensitivity analyses of print lot: (**a**) LC-CO$_2$; and (**b**) LC-NMVOC.

### 4.1.1. LC-CO$_2$

Water-based IJ, oil-based IJ, and oil-based GR printing all tended to have larger LC-CO$_2$ values as the printing lot became smaller. However, this tendency was remarkably large in oil-based GR printing and was relatively small in oil-based IJ. Therefore, it was confirmed that the effect of the print method is large, and the effect of the ink composition is small on LC-CO$_2$. As the printing lot became larger, the difference in LC-CO$_2$ due to the difference in printing method became smaller; however, the LC-CO$_2$ for GR did not drop below the LC-CO$_2$ for water-based IJ until about 670,000 sheets/lot. This was because the contributions from the ink loss and the loss of flexible packaging material were large and independent of the lot size for GR printing; therefore, their contribution to the environmental burden was dominant when the printing lot was small. In contrast, for inkjet printing, the power consumption in the printing step is larger than that in GR printing; for this reason, the LC-CO$_2$ for water-based IJ finally overtook the LC-CO$_2$ for GR. However, even beyond this inflection point, the advantage was small. Comparing a volume of 670,000 sheets/lot to monthly sales volumes across the spectrum of Kao products, for products such as cosmetics, which are manufactured in a wide variety but with a limited volume for each individual product, printing using water-based IJ has the clear advantage of reducing LC-CO$_2$. For fabric and home care products, which have more limited variety but are produced in large volumes, GR printing would show a slight advantage to reduce LC-CO$_2$. Volumes for the remaining product lines, including skin care and hair care products, with a wide range of product lines, fall near this inflection point. Our work shows that manufacturers can optimize their reduction of LC-CO$_2$ for package printing by selecting the printing technology based upon the lot size.

#### 4.1.2. LC-NMVOC

Considering the LC-NMVOC per sheet of flexible packaging, emissions for the water-based ink were much smaller than those for oil-based ink, regardless of the printing method and independent of the lot size. This was because the quantity of VOC in the aqueous ink composition is low, and the amount of ink applied is smaller than the amount of oil-based ink, due to the improvement of the pigment dispersibility in LUNAJET® used in the present study.

### 4.2. Sensitivity Analysis on Number of Ink Colors

In recent years, the demand for multi-color printing has increased, due to the diversification and complexity of printing designs. As inkjet printing systems print without contact while digitally controlling the quantities and proportions of inks, most of the color space can be reached using only the CMYK primary colors. In contrast, gravure printing applies ink through contact with the printing plate. In order to reach other colors in the color space using the CMYK primary colors on a gravure press, it is necessary to print and dry by hot air each primary color separately. In addition to this loss in efficiency, the imprecise alignment of the plates (misregistration) for the different primary colors can introduce errors in the print such as inter-color bleeding and a position misalignment in the overlapped patterns. For this reason, in gravure printing, complex images typically require special color inks, different from the CMYK primary colors.

For our study, LC-$CO_2$ and LC-NMVOC per sheet number were calculated as a function of the number of colors, up to eight colors, for a printed area of 2100 m$^2$, where the print area was the same, even if the total number of colors changed. The results are shown in Figure 8.

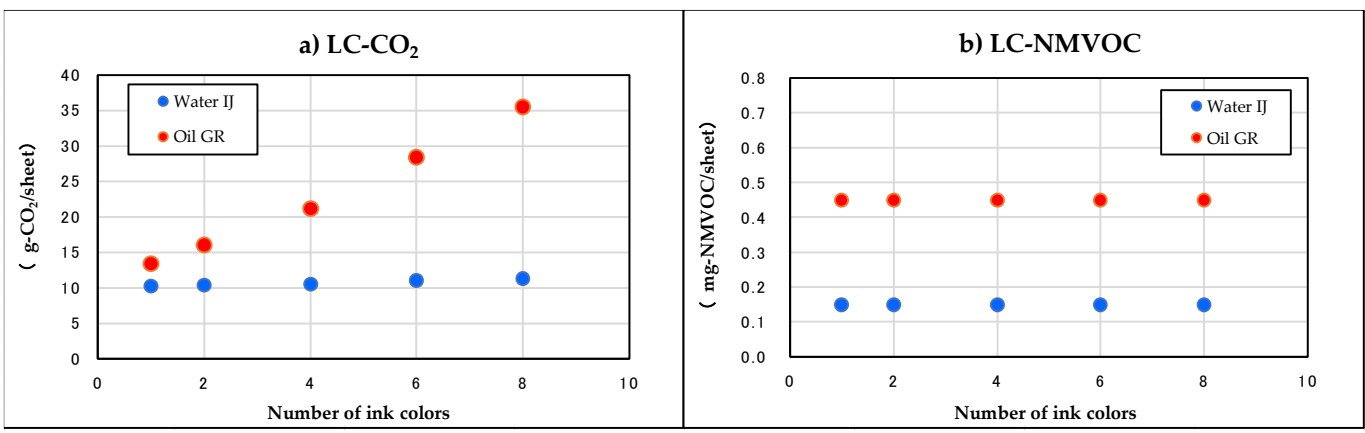

**Figure 8.** Sensitivity analyses on the number of ink colors: (**a**) LC-$CO_2$; and (**b**) LC-NMVOC.

#### 4.2.1. LC-$CO_2$ (2100 m$^2$ Printing, per Film Package)

As described above, for 2100 m$^2$ (small-lot) printing, the LC-$CO_2$ per sheet for Oil GR was much larger than that for Water IJ, due to the loss of ink and manufacture of the printing plates. As the number of colors increased, these effects were multiplied, as the ink loss applies to each color and each color requires plate manufacture. As the ink loss was small and printing plates were not required, the value was almost constant for water-based inkjet.

#### 4.2.2. LC-NMVOC (2100 m$^2$ Printing, per Film Package)

LC-NMVOC for both the Oil GR and the Water IJ showed a negligible change with an increase in the number of colors and, as before, was much smaller for Water IJ, in which the VOC fraction was low in the ink composition. Ink loss and printing plate manufacture, which created a large dependence on the number of colors for LC-$CO_2$, were considered

to have a small effect on VOC. Ink loss contained VOC but, in the present scenario, it is incinerated, such that it had an impact on $CO_2$ and a small impact on VOC.

### 4.3. Impact Assessment

The results of our environmental impact calculations for the four scenarios of Oil GR-L, Oil GR-S, Water IJ-L, and Water IJ-S were integrated over five environmental impact areas: Climate change, air pollution (urban areas), photochemical oxidants, fossil fuel consumption, and water consumption (groundwater). The inventory used for each environmental impact area is summarized in Table 4. LIME3 [47] was quoted for the integration assessment, the weighting factor was G20 (population-weighted), the input/output factor was Japanese, and the interest rate was 5%. The results of the integration (classified by life cycle stage, materials, and environmental impact) are shown in Figure 9.

**Table 4.** Environmental impact areas and inventory results used.

| Area of Environmental Impact | Inventory Used |
|---|---|
| Climate change | $CO_2$, $CH_4$, $N_2O$ |
| Air pollution (urban areas) | $SO_2$, $NO_X$, PM10 |
| Photochemical oxidants | $NO_X$, NMVOC |
| Resource consumption | Oil, Coal, NG |
| Water consumption | Water (groundwater) |

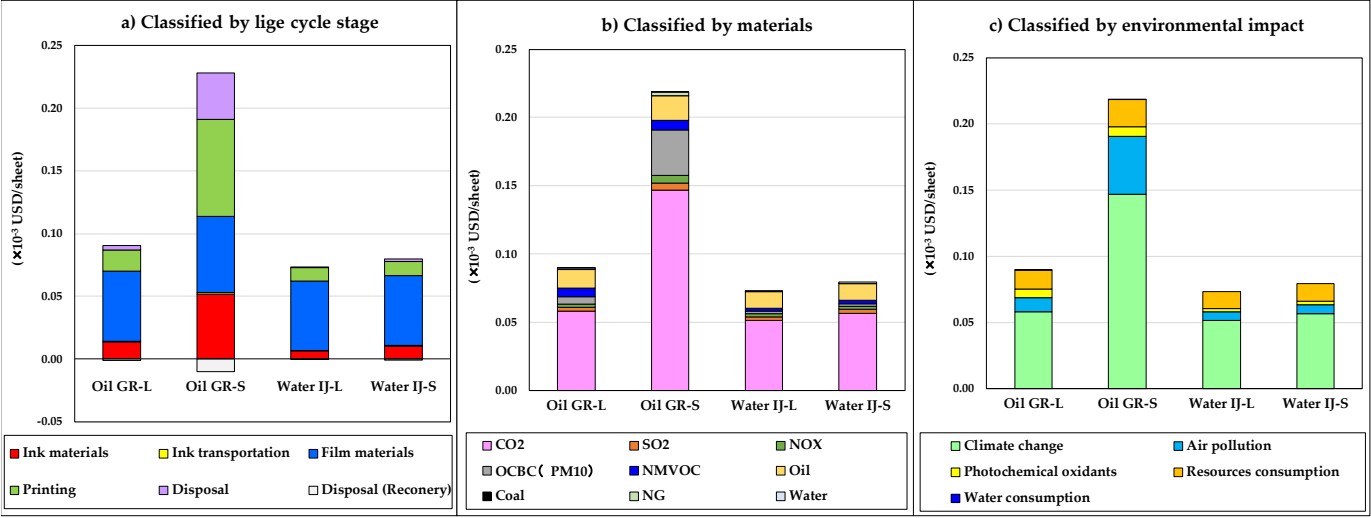

**Figure 9.** Impact assessment results (per printing process of film package). Classified by (**a**) life cycle stage; (**b**) material; and (**c**) environmental impact.

Classified by life cycle stage, the integration resulted in $9.0 \times 10^{-5}$, $2.2 \times 10^{-4}$, $7.3 \times 10^{-5}$, and $7.9 \times 10^{-5}$ USD per sheet for the Oil GR-L, Oil GR-S, Water IJ-L, and Water IJ-S scenarios, respectively. For the Water IJ, the effect of the printing lot size was small, while for the Oil GR, the cost per unit of the printing lot increased drastically when the printing lot was small.

Classified by materials, both scenarios were most impacted by $CO_2$, at 60–70% of the total, followed by oil (at 15–25%). NMVOC was not as high (at 2–6% of the total), but water-based IJs had about a third of the impact of oil-based GRs.

Classified by environmental impact, the effect of climate change was the largest, with 60–70% of the total, followed by resource consumption (18–28%), air pollution (about 8%), and photochemical oxidants (3–6%).

## 5. Conclusions

Previous reports on the environmental impact of printing have limited their evaluation to the consideration of ink composition or packaging material, and there have been few research examples of environmental impact evaluations considering the entire printing process. In this study, we compared the environmental performance of inkjet printing, when using the aqueous inkjet ink "LUNAJET®" (with image quality, stability, and drying improved by nanodispersion technology and precise control of ink surface tension), with oil-based gravure printing for the entire printing process, including ink procurement, materials procurement, printing, and disposal of refill flexible packages. The objective of this study was to clarify the impact of printing lots on the environment in both models, by evaluating the two scenarios of large-lot (21,000 $m^2$) and small-lot (2100 $m^2$) printing. The endpoints included inventory analysis, characterization, and impact evaluation.

Regarding the inventory analyses, LC-$CO_2$ did not show a significant difference between oil gravure printing and water inkjet printing in large lot printing; however, there was a significant increase in $CO_2$ emissions per unit in oil gravure printing for small-lot printing, while there was no significant change in water inkjet printing. In terms of LC-NMVOC, water inkjet printing was significantly smaller per unit, regardless of the lot size. These results reflect the difference between $CO_2$ emissions, in which the ink or material loss per unit changes depending upon the printing method, and the VOC emissions, which are dominated by the amount of VOC in the ink composition and the amount of ink applied per unit area.

With respect to the effect of the ink composition and the printing method on the environmental burden, it was confirmed that the effect of the difference of the printing method was large for LC-$CO_2$. The environmental burden of the aqueous inkjet method per unit was smaller than that of oil-based gravure printing for small print lots, while the environmental burden per unit of oil-based gravure printing method was smaller than that of the aqueous inkjet method per unit for large print lots. Through this study, it was confirmed that $CO_2$ emissions can be minimized by selecting the printing method according to the printing lot size. In addition, VOC emissions at printing sites for oil-based gravure printing increase not only the environmental burden but also the health risk of the work environment and the fire risk of the printing factory. It can also be stated that the cost of not only the environmental and safety burdens, but also the cost burden in terms of fixed costs, such as the health management cost of workers, the cost of construction of VOC recovery equipment, and the cost of installation of safety equipment, should be considered in the choice of printing technology.

A sensitivity analysis of the number of colors showed that, in terms of LC-$CO_2$, while $CO_2$ emissions increased as the number of colors increased in Oil-based GR, the effect upon Water IJ was negligible. This was because there is no need to manufacture plates for new colors when inkjet printing, and the amount of ink loss is also small; thus, the cost per unit is nearly independent of the number of colors for inkjet printing.

It has been confirmed that the environmental burden per unit for analog gravure printing can be reduced by making the print lot larger. This result can be applied to mass consumption products, for which the reduction of production costs is the highest priority. However, for products where packaging customization adds value, water-based IJ can deliver customized packaging with rapid turnaround and reduced waste, even for small lots, which are favored in modern e-commerce markets, without sacrificing a reduction of environmental burden.

The usefulness of aqueous inkjet printing for flexible packaging has been confirmed, but the problem remains that it is inferior, in terms of throughput and productivity, compared to mass-production gravure printing. In addition, the energy required for drying aqueous inkjet printing is higher than that needed for oil-based ink. These remain challenges for the future development of ink technology and the improvement of inkjet printing equipment. On top of that, it requires more studies in other regions since this study was performed only based on the market in Japan. However, the results of this study confirm

that investment in such developments would be worthwhile, as aqueous inkjet printing is an effective technology for environmentally friendly flexible package printing and the production of packaging materials with a smaller environmental burden.

**Author Contributions:** Conceptualization, N.I. and K.K.; methodology, N.I. and T.O.; validation, N.I.; formal analysis, N.I., T.O. and K.K.; investigation, K.K., T.E., H.K. and S.K.; resources, N.I.; data curation, N.I.; writing—original draft preparation, K.K., T.E. and T.O.; writing—review and editing, K.K., S.K. and T.O.; visualization, K.K., T.E., S.K. and T.O.; supervision, N.I. and Y.S.; project administration, N.I., K.K. and M.S. All authors have read and agreed to the published version of the manuscript.

**Funding:** This research received no external funding.

**Institutional Review Board Statement:** Not applicable.

**Informed Consent Statement:** Not applicable.

**Conflicts of Interest:** Some of the authors contributed other professional adviser are employed by the company of the studied system.

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
