# Peer review of "Environmental Impact Assessment of Flexible Package Printing with the “LUNAJET®” Aqueous Inkjet Ink Using Nanodispersion Technology"

_sustainability, doi:10.3390/su13179851_

Round 1
Reviewer 1 Report
- Please improve the abstract. It should provide compact and focused highlights of the study.
- Introduction lacks the citation especially, in first 2 paragraphs do not contain any citation. For instance, there should be appropriate references in support of following claims: (a) This has led to the geographic customization of packaging, as well as the shortening of shelf-lives for particular package designs , (b) ...... This has led to the geographic customization of packaging, as well as the shortening of shelf-lives for particular package designs , (c) Inkjet printing is a particularly promising printing method, as it has a printing speed comparable to conventional analog printing, as well as high color gamut, accessible through its unique image-forming process, (d) CO2 derived from raw materials used in the procurement of substrates accounts for over 60% of total emissions ....
- Please check the copy rights an other guidelines for Figure 1.
- Sec. 1.2.2. What are the milling conditions for reducing size of pigment powder to the nano-scale.
- Which electrostatic repelling groups is indicated by X+?
- Sec. 4.1.1. It is strange that effect of the print method is not influenced by ink composition. Please elaborate more the relevant discussion considering findings of other studies.
- Sec. 4.2. Please briefly describe drying conditions.
- How errors can be introduced due to the imprecise alignment of the plates.
Reviewer 2 Report
This is an article that fits within the scope of the journal. Good aspects of the work are: (1) it is well written and structured, (2) the studied product system is well detailed. However, the major issue is that the methodology lacks in clarity and transparency, and must better argued in certain cases. It is also needed to be more specific on certain matters (e.g. is this only for a case study in Japan?). Please see the specific comments. Hence, overall, I advise for a major revision.
Major comments:
A database is used that is representative for Japan. It should be made clear from the beginning and title that the focus is on evaluating this product in the Japanese context. Moreover, the authors should discuss its representativeness for the outcome for other regions.
It is unclear whether the studied product is the film printing or the flexible packaging. The title focuses on film printing and the functional unit as well, but then is mentioned in the introduction: “A complete assessment of the environmental impact of flexible plastic packaging re-quires consideration of the effects of their disposal after they are discarded”. Which one is it now? If I understand well it is printing on flexible packaging. Please also mention “flexible packaging” in the title.
Conflict of interest: The authors state no conflict of interest but some are apparently working for the companies of the products that is being studied. This is not an issue necessarily but please be explicit about this. Have other authors also been payed by the companies for making this work? Please be explicit about this.
If it is just printing, then is this independent or dependent on the type of material you print on? You use one particular flexible packaging. Is this representative for all?
Do the authors not mean “environmental impact assessment” instead of “environmental assessment?”
Specific comments:
Abstract:
The abstract lacks in a short description of the method (was LCA applied?) used and data sources (is this primary data?). This should be made apparent
Moreover, the abstract has a bit a long introduction. The focus of this work is not on the printer but on its environmental impact assessment.
Introduction:
The authors mention: “Our results show that these advantages can be realized while also decreasing the environmental bur-den, through the adoption of aqueous inkjet printing technologies.”. It is not common to discuss results or conclusions of your study in the introduction section.
The introduction lacks a good introduction on what LCA entails. General references on this methodology are also missing, e.g. ISO 14040-14044. The acronym is even not written in full the first time it is mentioned.
The authors mention: “For this reason, resource consumption was reduced by only 10% approximately and, in addition, a large amount of alcohol was released as a VOC into the atmosphere.” This is again a result, which does not belong in the introduction section.
The authors wrote: “Furthermore, environmental impact assessments of printing methods have been con-ducted in various fields, which have suggested that the CO2 derived from raw materials used in the procurement of substrates accounts for over 60% of total emissions in the print-ing of papers and films”. Please present (a) reference(s) for this statement.
The introduction contains a very elaborate explanation of the technology. It is interesting and it is a personal choice, but, in my opinion, this can be shortened to keep more a focus on the core aspect of the work: the environmental impact assessment. Yet, this is just an advice not an obligation.
- Material and methods
Be specific about the composition or technical type of the flexible packaging.
Figure 3. Please be consistent in either using boxes for processes and arrows for flows. Now it is mixed and difficult to comprehend
Figure 3. “landfill recycle” is mentioned, but is it now landfilling or recycling? Please be clear on this matter.
Be specific about the location (Japan and city if possible) and relate this with the selection of the database and specificity of the inventories.
2.2 Raw data collection
A specification and overview of all the foreground data is missing. A table would be helpful
Specify all the names of the processes/flows that have been used from the database, for the sake of transparency and reproducibility.
Specify what you mean with “fuel oil A equivalent”
It is not clearly described how waste valorization was handled. If I understood well, the avoided burden approach was used, since impact is substracted. Yet, is this also the case for recycling? And how was reuse exactly handled? (if avoided burden is applied for heat recovery than it should also be applied for recycling) Please also argue your choices and whether or not they align with allocation rules of the selected database.
Moreover, is there no electricity generation during burning? Are you sure the heat is actually used for heating buildings and not just dissipated? What is the displaced heat mix? The fuel oil equivalent A? Why was this one picked?
The authors mention: “In addition, VOCs (volatile organic compounds) discharged by ink dry-ing at the time of printing were taken as atmospheric releases” It is unclear what now the data source is? Measurements?
It is unclear whether printers themselves are included in the inventory or not? Is a different printer not used in the scenarios? Should this then not be covered?
2.3
Specify the versions of the used impact methods
Specify the exact reference for the AR5 coefficients.
- Results
You mention: “and most of the NOx was the effect of NMVOC”. Please explain this better
Round 2
Reviewer 2 Report
Overall the authors have addressed some comments in an adequate manner, but some not. A second revision is therefore needed. Please see below comments.
Concerning conflict of interest, please read the author guidelines. In the author guidelines is mentioned: “All authors must disclose all relationships or interests that could inappropriately influence or bias their work. Examples of potential conflicts of interest include but are not limited to financial interests (such as membership, employment, consultancies, stocks/shares ownership, honoraria, grants or other funding, paid expert testimonies and patent-licensing arrangements) and non-financial interests (such as personal or professional relationships, affiliations, personal beliefs).” Since some of you authors are affiliated with the Kao Corporation, you should specify this. For example, mention in a conflict of interests section: “Some of the authors are employed by the company of the studied system”.
Concerning the material type, please present your statement in the manuscript. Do you have a reference that justifies the representativeness of polypropylene? (don’t just reply to me as a reviewer; I present comments so you could improve the manuscript; please adapt the manuscript)
In the abstract is written: “ on endpoint modeling 3)”. Why do you mention “3”?
Concerning the specification of the flexible packaging, please be more specific about what the composition of “Rakura Ecopack” is. Which type of plastic is this? How representative is it for the Japanese market?
Regarding the inventory data, I mean the raw primary inventory data, not the compiled LCI inventory data. Raw primary data would for example be the amount of electricity production etc.
Please be specific about what you mean with “fuel oil A”
My comment on the methodological explanation and justification of waste valorization has not been addressed. Please see literature references on this topic (Schaubroeck et al., 2021; Schrijvers et al., 2016a, 2016b). Please be specific about your approach. Is this substitution/avoided burden?
Please present in the manuscript that 100% energy recovery is an assumption.
The authors mention in the reply: “This is just a calculation based on our assumption. Neither from data nor measurements”. Be explicit about this in the manuscript and mention the calculation.
Concerning the inclusion of printers, discuss and justify why these are not included in the study.
Schaubroeck, T., Gibon, T., Igos, E., Benetto, E., 2021. Sustainability assessment of circular economy over time: Modelling of finite and variable loops & impact distribution among related products. Resour. Conserv. Recycl. 168, 105319. https://doi.org/10.1016/j.resconrec.2020.105319
Schrijvers, D.L., Loubet, P., Sonnemann, G., 2016a. Critical review of guidelines against a systematic framework with regard to consistency on allocation procedures for recycling in LCA. Int. J. Life Cycle Assess. 1–15. https://doi.org/10.1007/s11367-016-1069-x
Schrijvers, D.L., Loubet, P., Sonnemann, G., 2016b. Developing a systematic framework for consistent allocation in LCA. Int. J. Life Cycle Assess. 1–18. https://doi.org/10.1007/s11367-016-1063-3
Round 3
Reviewer 2 Report
The authors have addressed the majority of the comments adequately. One comment should be better addressed, namely concerning the exclusion of the printer device itself. Are entirely different printers used? Would it have a huge impact on the result? (how much film can be printed per printer?) Excluding the device production and disposal seems a limitation that needs to be discussed in the discussion section. Please address this minor comment.
Author Response
The authors have addressed the majority of the comments adequately. One comment should be better addressed, namely concerning the exclusion of the printer device itself. Are entirely different printers used? Would it have a huge impact on the result? (how much film can be printed per printer?) Excluding the device production and disposal seems a limitation that needs to be discussed in the discussion section. Please address this minor comment.
Response :
Thank you for the comment. We didn't consider the point in this study. So we added an explanation in Figure3 "The environmental loads of the printer device production were not included into the calculation boundary in this study."
Indeed, the point you mentioned would give a certain impact on the study, but we don't think it is huge because the printers can be used more than 10 years, which can dilute the impact. If we print 20000m/day and run 300days/year, it would mean at least 60000km in a life of printer. What we are discussing here is like a scenario of 2-20km. Therefore we suppose the impact would be small. But we will diffenitately consider your point in next studies, thank you very much.